# Subjective Complaints and Coping Strategies of Individuals with Reported Low-Frequency Noise Perceptions

**DOI:** 10.3390/jcm13040935

**Published:** 2024-02-06

**Authors:** Kristina H. Erdelyi, Anselm B. M. Fuermaier, Lara Tucha, Oliver Tucha, Janneke Koerts

**Affiliations:** 1Department of Clinical and Developmental Neuropsychology, Faculty of Behavioral and Social Sciences, University of Groningen, 9712 TS Groningen, The Netherlands; k.h.erdelyi@rug.nl (K.H.E.);; 2Department of Psychiatry and Psychotherapy, University Medical Center Rostock, 18147 Rostock, Germany; 3Department of Psychology, National University of Ireland, W23 F2H6 Maynooth, Ireland

**Keywords:** low-frequency noise, LFN, complaints, cognition, depressive symptoms, sleep, fatigue, stress, coping

## Abstract

**Background**: Subjective everyday hindrances associated with low-frequency noise (LFN) can be high; however, there is still a lot unknown about experienced complaints. This study aims to investigate (1) subjective complaints and (2) coping strategies of individuals reporting everyday hindrances from LFN. **Methods**: Cognition, depressive symptoms, sleeping, fatigue, stress, and coping questionnaires were administered to participants sampled for their LFN complaints (LFN1 = 181), LFN complainants derived from a community sample (LFN2 = 239), and a comparison group without LFN complaints (CG = 410). **Results**: Individuals reporting LFN perceptions reported complaints in all domains and showed a higher proportion of above average symptom severity compared to the CG. Most complaints were reported by the LFN1 group, the least by the CG. However, on some sleeping, fatigue, and stress-related variables, a similar or even higher symptom severity was observed in the LFN2 group. Further, all groups used a similar combination of multiple coping strategies, although the LFN1 group scored higher on support seeking. **Conclusions**: There might be differences in the complaint severity between different LFN subgroups and future investigations of primary and secondary complaints are necessary. Also, more research about the use and success of coping strategies for LFN-related hindrances are needed for clear conclusions.

## 1. Introduction

### 1.1. Low-Frequency Noise (LFN)

While the effects of noise pollution are widely researched and its various adverse effects are recognized [1,2,3], there is still a lot unknown about LFN. LFN is defined as noise at low frequencies between 20 and 100/125 Hz by the Dutch Institute for Public Health [4]. Sounds below 20 Hz are defined as infrasound [4]. However, definitions can vary and some definitions of LFN encompass wider frequency ranges (e.g., between 10 Hz and 200 Hz in [5]). The primary sources of LFN are man-made, such as traffic, ventilation, or household and industrial installations. The rapid growth of industrialization is also accompanied by a rising number of concerns and LFN-related complaints [4,6,7]. LFN is most commonly perceived by hearing a deep humming, rumbling, or engine-like sound, but bodily vibrations or other kinds of non-auditory perceptions are also reported [5,6,8]. LFN is only perceived consciously by a proportion of the general population; however, it is not yet clear how big this proportion is and why some individuals report LFN perceptions and others do not. Prevalence estimations for the proportion of the general population perceiving LFN in various studies range between 2% to up to 34% with an estimated pooled prevalence of 10.5% [9]. Moreover, it is not yet clear why some individuals seem to be more annoyed by LFN than others. In terms of experienced annoyance, the Dutch Institute for Public Health estimates that 2% of the Dutch adult population experiences severe annoyance, and 8% experiences some annoyance from LFN [4]. Complaints reported in relation with LFN perceptions can be manifold and can have considerable effects on the daily functioning and health of those reporting LFN perceptions. They can encompass various physical (e.g., cardiovascular complaints, or nausea), psychological (e.g., annoyance, stress, sleeping problems), cognitive (e.g., concentration difficulties), or social (e.g., work incapacity, relationship problems) domains [5,6,8,10]. However, more systematic research has to be conducted on which complaints occur most frequently, on which complaints can be directly associated with LFN exposure, on influencing (non-acoustic) factors, and on possible primary or secondary complaints. Also, it is still unclear to what extent objectively measured LFN exposure aligns with subjectively measured LFN perceptions. Previous research has utilized these two concepts to different extents. The present study focuses on a systematic investigation of complaints in the cognitive and psychological domains in relation to subjective LFN perceptions.

### 1.2. Cognitive Complaints Reported in Relation to LFN

Cognitive functions, including attention, memory, and executive functions, are necessary for activities of daily life. They are also among the most commonly reported LFN-related complaints, especially concentration difficulties [5,6,8,10,11]. Difficulties can occur in so-called attention functions, such as the selection or awareness of specific stimuli while disregarding interruptive stimuli or the regulation of the intensity of awareness [12]. Subjective reports of concentration difficulties in surveys in relation to LFN ranged from 7.5 to 17% [13], over 43 to 44% [6], and up to 67% [8] of research participants. Still, research trying to objectify subjective attention difficulties shows incongruous results. While some studies suggested worse attention performance during LFN exposure [10,14,15,16], other studies observed better or similar performance during LFN exposure [17,18,19].

When looking at memory, i.e., the ability to encode, retain, retrieve, and reactivate information [20], there is to our knowledge hardly any research aiming to explore subjective or objectively measured memory complaints in relation to LFN. In a treatment study [21], on average participants with reported LFN perceptions stated before treatment that the noise leads them to forget things some of the time (answer ‘2’ on a 5-point scale from ‘0 = not at all’ to ‘4 = most of the time’). Considering objective memory measurements, two studies suggested worse memory functioning [14,19], of which one has also found some memory variables with no relation between LFN exposure and memory [19].

A third group of cognitive functions are executive functions, a group of complex, higher-order functions especially relevant for cognitively demanding tasks. These metacognitive processes generate, synchronize, coordinate, or withhold activity and involve processes like planning, working memory, the inhibition of responses, or the ability to shift quickly between tasks [22,23]. There is to our knowledge no research specifically aiming at investigating subjective executive functions in relation to LFN. Studies using objective performance tests focused mainly on inhibition and working memory. Again, some studies showed worse performance [15,16,24,25], while others suggested better [17,18,26], or no differences in performance [10,27] related to LFN exposure. Notably, a recent meta-analysis suggests that a negative impact of LFN exposure on cognition is observed only on higher-order cognitive functions, such as logical reasoning, mathematical calculation, and data processing, compared to basic functions in the areas of attention, memory, and executive functions [28].

In conclusion, there is still a lot unknown about the effects of LFN on cognition, especially in terms of memory and executive functions. Evidence of subjective complaints is currently stronger compared to objectively measured cognitive functioning, which does not allow for firm conclusions, yet [29]. Subjective complaints have mainly been investigated in terms of attention functions, although these investigations were often conducted using single items or open questions.

### 1.3. Psychological Complaints Reported in Relation to LFN

#### 1.3.1. Depression

The impact of mental disorders on quality of life and years lived with disability can be high. Depression currently ranks as one of the most common and most impactful disorders [30]. Although many of the reported complaints from LFN are also complaints associated with depression (such as depressed mood, sleeping problems, fatigue, concentration difficulties), research investigating the association between LFN and depression is scarce and provides conflicting results. Indications for increased depressive symptoms were found in a questionnaire study, which observed that 30% of LFN complainants showed moderately severe and severe depression symptom severity compared to 5% of a matched comparison group living in the same building block [31]. Further, in a treatment study, 53% of the participants reported that LFN made them feel depressed before treatment [32]. In a follow-up study [21], participants with reported LFN perceptions stated on average that the noise made them feel depressed some of the time (answer ‘2’ on a 5-point scale from ‘0 = not at all’ to ‘4 = most of the time’) before treatment. 

However, other studies provide conflicting results. Road traffic noise exposure in the low-frequency ranges was not found to correlate with a diagnosis of depression [33]. Also, research on wind turbine noise, a possible source of LFN, did not find correlations between depressive symptoms and noise exposure [34]. Finally, similar proportions of participants who reported experiencing depression living in areas of high (19%) and low (18%) assumed LFN exposure were observed in [35]. In conclusion, evidence of subjective reports of depression or depressed mood in association with LFN seems to be stronger compared to research trying to link reports of depression to LFN exposure. However, the research using subjective LFN reports has to be reviewed carefully due to partly unstandardized measures or small sample sizes. With the high impact of depression on everyday life, a standardized investigation of depressive symptoms in individuals reporting to be affected by LFN in their daily life is necessary.

#### 1.3.2. Sleep

Among the most commonly reported and also the longest and most extensively studied complaints are sleeping problems and disturbed rest. Various case, pilot, field, survey, and experimental studies, as well as reviews, investigated the role of LFN or of noise with LFN components on sleep, e.g., [4,9,11,21,32,36,37]. In terms of subjective complaints, the proportion of individuals reporting sleeping problems related to LFN in surveys ranged from 13 to 22% [13], over 54 to 77% [8], 82 to 89% [31], and up to 83% [6]. Further, significantly higher sleep disturbances, difficulties falling asleep, and tension in the morning were reported from individuals annoyed by LFN compared to individuals not annoyed by LFN [13]. 

Research utilizing both subjective questionnaires and objective measures of sleep shows contradicting findings. From two similar sleep laboratory studies with nocturnal LFN exposure and cortisol measurements [38,39], only one found altered, flat cortisol levels 30 min after awakening [38]. This was related to lower sleep quality and negative mood. Further, this research observed longer reported times to fall asleep [38], which was also not found in the follow-up study [39]. In this latter study [39], higher tiredness in the morning, lower social orientation, and negatively affected mood were observed in relation to LFN [39], but no change in the number of nocturnal sleep disruptions or morning tension was observed. In a different field study with cortisol measurements, increased night cortisol levels were observed in the first half of the night in children exposed to traffic noise with LFN components [14], which could be associated with decreased sleep quality. Further, higher noise levels in this study were related to more subjective sleeping problems. Contradictory findings were also observed in a study using actigraphs [40], where participants reported worse sleep quality during LFN exposure; however, objective measurements indicated better sleep. Finally, an experimental EEG study did not find significant sleep differences between nights with and without LFN exposure [41]. 

In conclusion, evidence of subjective sleeping difficulties is currently stronger compared to objectively measured sleep disturbances. Further, studies differ regarding which aspects of sleep seem to be affected. Difficulties might thus occur in only some areas of sleep or might differ between subgroups and situations. Notably, previous studies had methodological limitations, such as single-item questions, small effect sizes, unclear noise dose descriptions, or the use of specific subgroups. Considering that good undisturbed sleep is necessary for maintaining good health and daily performance [42], a deeper understanding of sleep and its specific subcomponents in individuals with daily life LFN perceptions is crucial.

#### 1.3.3. Fatigue

Although LFN exposure has been commonly associated with sleeping problems, it has also been suggested to induce short periods of sleep [5]. Especially in work conditions, LFN was associated with increased drowsiness, short-term tiredness, and (mental) fatigue ([43,44], see overviews in [5,45]). Further indications for an association between LFN and fatigue complaints can be derived from studies with LFN exposure during mental performance tasks [24,27,46]. Specifically, higher reported mental fatigue was observed with higher LFN levels and with rising cognitive workloads [24]. Further, significant correlations between self-rated tiredness and annoyance due to LFN, impaired working capacity, and response times were found [27]. Notably, higher rated tiredness during a long series of performance tasks was found in relation to both LFN and also non-LFN ventilation noise [46]. Further, sleep laboratory studies [5,38,39] give first indications for an association between LFN and fatigue. A series of laboratory EEG studies by Landström and colleagues (see [5] for an overview) associated ventilation noise with LFN tones with greater fatigue. Also, more tiredness after nights of LFN exposure was observed by [38,39]; however, this only reached significance in one of those studies. While much of the previous research focused on short-term fatigue, little research has considered long-term fatigue reflecting a status of tiredness and diminished functioning [47] in the daily living of affected individuals. Two surveys suggested proportions of individuals reporting fatigue in relation to LFN of 56% [6] and 59% [31]. However, in another survey [8], fatigue did not belong to the five most commonly reported complaints and was included as one of many secondary effects in the ‘other’ category encompassing 39% of the participants. Finally, individuals annoyed by LFN showed significantly higher rated levels of fatigue compared to non-annoyed individuals [13].

Overall, the current findings give indications, but do not allow for clear conclusions on the relation between LFN and everyday fatigue. This might partly be due to the difficulty with defining fatigue and distinguishing it from symptoms of tiredness or drowsiness. Considering the importance of fatigue on daily performance, further research on the frequency, severity, and impact of long-term fatigue in daily life is crucial in individuals with daily LFN perceptions.

#### 1.3.4. Stress

Constant LFN has been classified as a background stressor and has been associated with various stress-related complaints, as also described in the previous paragraphs [5,48]. Considering self-reported stress, a proportion of 57% of respondents reported feeling stressed in relation to LFN in [6], and in a small-scale treatment study, 56% of the participants reported feeling distressed due to the noise before treatment [32]. Further, in a follow-up treatment study [21], 31% of the participants were classified as highly stressed, and 41% were classified as moderately stressed before treatment. Contrarily, stress was not among the five most common complaints in the survey by Moller and Lydolf [8], and stress was included in the ‘other’ category with other complaints (encompassing 39% of the participants).

Both self-reported stress and stress measured by cortisol levels, a hormone that can be disrupted by stressful events and which shows elevated levels during short-term stress, were included in a task–performance study by [49]. Participants were categorized as having high or low sensitivity to LFN and general noise, and were exposed to LFN and non-LFN. The results suggested that the LFN-sensitive participants reported the highest stress levels during all cognitive tasks and all noise conditions. Also, higher stress levels after LFN exposure were related to participants not feeling in control, regardless of their sensitivity. Further, a small but significant cortisol elevation level during LFN exposure was observed for individuals who were sensitive to noise in general, but not specifically in LFN-sensitive individuals. Surprisingly, these measured cortisol levels did not clearly correlate with the subjective ratings of stress. Another surprising finding was that non-LFN was in general reported to be more stressful than LFN by all participants. These findings suggest that the relationship between LFN exposure, noise sensitivity, and reported short-term stress is not clear yet, and indicate a discrepancy between subjective and objective stress measures. 

Since the previous results of subjective stress related to LFN are partly based on single-item questions, and since objective measures have focused on short-term stress, a thorough investigation of everyday stress in individuals with daily LFN perceptions is needed. Moreover, other factors, such as sleep, seem to be influential on experienced stress and need to be considered (as shown in LFN nonspecific research [50] or as indicated by altered cortisol levels observed in the previously described sleep studies [14,38]). 

### 1.4. Coping

Coping is a multidimensional term with various interpretations. In social and behavioral sciences, coping refers to the cognitive and behavioral strategies used to manage demanding or stressful situations [51]. Such strategies can be subdivided into different categories. Two major ones are ‘problem-focused coping’, strategies focusing on changing the source of stress, and ‘emotion-focused coping’, strategies addressing the reduction of one’s emotional distress [52]. There are also further theoretical frameworks, such as ‘active problem-oriented coping’, which involve active techniques to avoid or reduce stressors, ‘support seeking’, which involve seeking assistance, advice, or understanding, and ‘avoidance behavior’, which involve actions aiming to escape or disengage with the stressor and its effects [52]. Determining which coping strategies prove to be useful does not only depend on the use of specific strategies, but can also change with the extent of its use or the context. The same coping strategy can be useful in one situation, but not in another situation.

An investigation of the application and success of coping strategies is especially relevant for LFN-associated complaints. First, LFN presents with properties that can make coping more difficult compared to general environmental noise. For example, the source of LFN can often not be found and individuals can have difficulties localizing or defining their LFN perception. This can lead to difficulties with identifying an external stressor. Second, compared to general environmental noise, LFN is usually more invasive, and insulation from the noise is less effective. This can make it complicated to resolve, reduce, or escape the LFN. While LFN seems to be reported as even more annoying than regular noise, others often do not share the same perceptions, and receiving (social) support can thus be challenging for affected individuals. All these properties can make it difficult to find successful coping strategies for the complainants and thus LFN can become a long-term stressor [4,5,36,53]. Notably, environmental noise research suggests that noise exposure can only partly explain noise reactions (such as annoyance). Other non-acoustical factors including personal, contextual, and noise management-related factors also play a highly relevant role in noise reactions and health effects [54,55,56]. Relevant factors include noise sensitivity, demographic factors, situational and personal circumstances, perceived control, and also coping mechanisms [5,56,57,58,59].

Despite the numerous concerns about LFN-related complaints, to our knowledge, coping, as earlier defined, has not been the main focus of previous research. However, studies assessing the ways in which affected individuals react to the noise and treatment studies give some first indications. Specifically, it seems that the majority of individuals reporting LFN perceptions try numerous actions to reduce nuisance, thereby often focusing on the external termination of the noise (e.g., using earplugs, masking the sound, switching off suspected sources, or changing or adapting their living location). However, these actions were regularly reported as unsuccessful in reducing nuisance [6,8,60]. Thus, personal or internally focused actions could be promising for reducing LFN-related nuisances. Such actions could be distraction [60] or strategies focusing on living with the sound and managing the experienced hindrances. In a pilot study, the use of electronically emitted masking sounds was effective at reducing complaint severity for most LFN complainants [60]. The authors concluded that this reduction partly stems from participants ‘learning to live with the sound’, and hypothesized multiple underlying mechanisms for the masking effect, including psychological distraction. Further, four treatment studies investigated the success of treatment procedures to reduce LFN-related complaints. Interventions that seemed to improve the quality of life and quality of coping included the (self-help) coping techniques of different relaxation therapies, [32,61], Neural Linguistic Programming/Visual-Kinaesthetic Dissociation [32], imaginal exposure [32], anchoring [32], cognitive behavioral therapy that focuses, amongst other things, on healthy thinking about sounds [21], desensitization for noise-related stress [21], and sound therapy [61]. Interventions were partly applied on-site and partly at home online or via electronic instructions. Evaluations differed individually regarding what techniques were especially helpful, and interventions were beneficial for some, but not all participants. Interestingly, two studies noted that participants had difficulties with coping, avoided noisy situations, missed out on things they like, and worried about the noise, their future coping abilities, and about making other people feel uncomfortable [21,32]. Notably, the proportion of individuals worried about the noise increased after the therapy sessions (from 67% to 78%) in one study [32]. Therefore, a high need for personal coping strategies targeting, amongst other things, the affected individual’s need to regain control over their personal environments, was recommended. 

In summary, current research indicates that affected individuals might commonly apply active problem-oriented strategies, especially focusing on external noise sources with mixed success rates. However, with no clearly identifiable external source or non-escapable LFN perceptions, emotion-focused strategies might be promising in reducing LFN-related hindrances for some individuals. However, the results of the treatment studies have to be viewed with care due to partly small sample sizes, the simultaneous use of multiple interventions that limit the identification of their individual effects, and the use of measures not specifically validated for assessing coping mechanisms. While support seeking strategies have not been specifically investigated in previous research, they have been named as a promising coping component for LFN complaints [32]. Thus, considering that the properties of LFN make successful coping especially difficult, and that there is a large variety of complaints reported by individuals with LFN perceptions, more insight into coping mechanisms is crucial. 

### 1.5. Research Aims and Questions

There are currently indications that daily life LFN perceptions could be associated with complaints in the domains of cognition, depression, sleep, fatigue, and stress. However, previous research presents mixed results and is limited. Therefore, this study aims to, first, investigate those subjective complaints, including complaints associated with cognitive functioning, such as attention, memory, and executive function, depressive symptom severity, different aspects of sleeping difficulties, fatigue symptom severity, and daily stress, using standardized and validated measurement instruments. Furthermore, there is a high need for investigating the coping strategies used in dealing with those complaints. Therefore, the second aim of this study is to provide a multi-faceted investigation of different coping mechanisms applied by LFN complainants. For this, an observational, cross-sectional questionnaire study was conducted comparing complaints and coping strategies between a sample of individuals experiencing LFN-related complaints in their daily life and individuals with no LFN-related complaints. This research is the first, to our knowledge, to integrate all those complaints in one study and to provide cross-comparisons between measures in contrast to previous unidimensional research including only one/a few of those domains. This also allows for a comparison between the frequency and extent of multiple complaint domains. Further, this research makes use of large naturalistic groups and an extensive battery of validated measures with big normative groups for better clinical interpretation. Eventually, the following research questions are proposed: What types and severity of complaints do individuals with LFN perceptions report?Do individuals with LFN perceptions report more complaints compared to individuals with no LFN perceptions?What proportion of individuals with and without LFN perceptions report complaints in a presumably clinically relevant range?What kind of coping mechanisms do individuals with LFN perceptions use compared to individuals with no LFN perceptions?

## 2. Methods

### 2.1. Procedure and Participants

LFN participants specifically sampled for LFN-related experiences (LFN1 group) were recruited via an online information letter that was distributed by the Stichting Laagfrequentgeluid (www.laagfrequentgeluid.nl, accessed on 11 December 2023), a Dutch volunteer organization with the goal to inform about LFN and to support affected individuals. Interested participants indicated their intention to participate via email. In order to participate, individuals had to be at least 18 years old, have good command of Dutch, and have current LFN perceptions and LFN-related difficulties in their daily life. Having current LFN perceptions was based on participants’ subjective reports and did not depend on a successful sound measurement. Participants received a definition of LFN formulated by the Dutch Institute for Public Health and Environment [4] in an information letter. Further, LFN-related complaints were assessed, first, by asking how often participants experience LFN-related complaints on a 5-point rating scale from 0 ‘never’ to 4 ‘continuously’. Second, participants had to rate the extent of LFN-related hindrances in their daily life between 1 ‘not at all’ and 10 ‘very much’. After signing up for the research, participants received a paper–pencil version of a battery of questionnaires together with an informed consent form and a form to sign up for a further LFN study. No financial reward was provided. A response rate of 65% was observed with 306 initially interested participants and 200 received questionnaires that arrived between June 2018 and February 2021. Usually, no reason was provided for not returning questionnaires. However, some initially interested participants provided reasons, including the hope for an on-site sound measurement instead of completing questionnaires, or time constraints. From those 200 questionnaires, seventeen participants were excluded for not fulfilling the inclusion and exclusion criteria, and two for having too many missing values (less than 15% filled out). Furthermore, six individuals were not retained in the LFN1 group since they reported LFN complaints to occur ‘never’, or only ‘sometimes’, in addition to a very low extent of hindrance (reported as a one or two). One person was excluded for not providing an answer to these questions. Finally, participants with a significant self-reported psychiatric or neurological disorder (e.g., schizophrenia or epilepsy) that could have confounding effects on sound perceptions or LFN-related psychological and cognitive functioning were excluded (*n* = 5). Individuals with psychiatric (*n* = 32, 18%) or neurological disorders (*n* = 6, 3%) with assumed low confounding effects on the outcome variables were not excluded (e.g., depression). Individuals with a diagnosis of tinnitus (*n* = 37, 20%) could participate in the research, since LFN perceptions and tinnitus can be comorbid, and since it is difficult to separate tinnitus patients from LFN-perceiving individuals [5]. Eventually, the final LFN1 group consisted of 179 individuals living in The Netherlands (99%) and two in Belgium (1%). The sex, age, educational level, and marital status of the LFN1 and all comparison groups, as well as the frequency of complaints and extent of hindrances, are depicted in Table 1. Further information about occupational status, living situation, and experienced LFN perceptions can be found in Erdélyi et al. [62].

A comparison group was recruited through “PanelInzicht”, a Dutch online research panel gathering research participants from the general public for financial compensation. These participants recruited via ‘PanelInzicht’ consisted of Dutch adults that have good command of Dutch and with similar demographic distributions regarding sex, age categories, and educational categories to the LFN1 group. These categorizations are described in detail in Erdélyi et al., 2023 [62]. Individuals with any self-reported psychiatric or neurological disorder or a diagnosis of tinnitus were not eligible for participation and three participants were excluded due to presumably invalid answer patterns. Since a large number of individuals in this group also reported LFN experiences, this sample was subsequently divided into a group of individuals with LFN-related complaints (LFN2 group) and a comparison group (CG) with (almost) no LFN-related complaints. For this division, the frequency of LFN-related complaints and the extent of their hindrances in daily life were utilized after participants received a definition of LFN. Individuals reporting complaints ‘regularly’, ‘often’, or ‘continuously’ and/or individuals reporting the extent of hindrances as a score of three or higher formed the LFN2 group. Thus, this LFN group has not been sampled specifically for their LFN complaints. Participants that reported complaints to occur ‘never’ or ‘sometimes’ and also reported the extent of hindrances in their daily life as a score of one or two, were retained as the comparison group (CG). Twelve participants were excluded, since they did not provide an answer to whether they experience LFN-related complaints or hindrances. 

The final LFN2 group consisted of 239 participants living in The Netherlands (97%) and eight in Belgium (3%). The final CG consisted of 468 participants living in The Netherlands (98.5%), six in Belgium (1.3%), and one in Ireland (0.2%). Notably though, the LFN1 group showed a significantly higher frequency of complaints with a large effect size (z = −13.87, *p* = <0.001, r = −0.68) compared to the LFN2 group (Mdn LFN1 = ‘often’, LFN2 = ‘sometimes’), and a significantly higher extent of hindrance with a large effect size (z = −11.17, *p* = <0.001, r = −0.55) compared to the LFN2 group (Mdn LFN1 = 8, LFN2 = 4).

Because the full questionnaire was too extensive for a regular study at the research panel (about 90 min fill-out time), two subsamples of similar demographic characteristics had to be obtained. Subsample A (SA) filled out the first half of the questionnaire battery (including measures of cognitive functioning and depressive symptoms), and subsample B (SB) filled out the second half (including measures of sleep, fatigue, stress, and coping). This led to the following samples: First, a LFN group that was specifically sampled for their LFN-related complaints (LFN1 = 181). Second, a LFN group that was not specifically sampled for their LFN-related complaints (LFN2 = 239), which was subdivided into LFN-SA = 131 and LFN-SB = 108 based on the individuals filling out the first and the second half of the questionnaire. Third, a comparison group with individuals with no LFN-related complaints (CG = 468), which was subdivided into CG-SA = 229 and CG-SB = 239 based on the individuals filling out the first and second half of the questionnaire. 

Due to exclusions and the division into a LFN2 group and CG, the distribution of some demographic characteristics differed significantly (Table 1). Specifically, the LFN1 group presented with significantly more low educated individuals compared to all other groups. Further, the LFN1 and LFN2-SB groups presented with significantly less married individuals compared to most other groups, and the LFN2-SB and CG-SA groups presented with significantly more widowed individuals compared to most other groups. Finally, the comparison groups were significantly older than all other groups. For more details, please refer to Appendix A. 

All of those significant differences were, however, of a small effect size (based on Cramer’s V and Cohen’s r, see Appendix A). Only one difference approached a medium effect size (age between the LFN2-SA and CG-SA groups). The potential influence of age, education, and marital status on the outcome variables was investigated. A correlational analysis between age and all outcome variables showed correlations of small size and most correlations demonstrated explained variances (based on R^2^) between less than 1% and 4%. The effect of low education level on all outcome variables, as investigated with Mann–Whitney U tests, only showed significant differences on two variables measuring stress and coping with small effect sizes. Finally, the effect of marital status (i.e., being married, unmarried, or widowed) was investigated on all outcome variables with Mann–Whitney U tests. Although significant differences were observed between all three marital statuses on at least one variable for the domains of cognition, depressive symptoms, sleep, fatigue, stress, and coping, all differences were of a small effect size. In conclusion, the effect of the observed demographic differences between the groups was assumed to be negligible. Further, the main reason for recruiting similar comparison groups was that LFN complainants in various survey studies [5,6,8] and in our LFN1 group present with older, more female, and more highly educated individuals compared to the general population (i.e., Dutch adult population presenting with an average of 50 years, 51% females, 24% low, 30% middle, and 28% high educated individuals; see [62]). Thus, our comparison group still clearly differs from the demographic characteristics of the general population and resembles the LFN1 group. Accordingly, the five groups were retained for further analysis. 

This study is embedded in a large-scale LFN research project at the University of Groningen. Ethical approval for the research project was obtained from the Ethical Committee of Psychology (ECP) affiliated with the University of Groningen, The Netherlands (Registry nr. 17255, PSY-1819-S-0165, PSY-2122-S-004).

### 2.2. Materials

This study is part of a larger research project and investigates cognition, depressive symptoms, sleep, fatigue, stress, and coping. Only these self-report questionnaires are described here. Results of other parts of the research project are published in Erdélyi et al. [62]. Questionnaires were chosen if they were proven to be clinically sensitive at detecting complaints and if they were psychometrically studied and validated. The LFN1 group completed all questionnaires. The LFN2-SA and CG-SA groups filled out the questionnaires measuring cognition and depressive symptoms and the LFN2-SB and CG-SB groups completed the questionnaires measuring sleep, fatigue, stress, and coping. 

Cognitive functioning was first measured with the 35-item Questionnaire for Complaints of Cognitive Disturbances (FLei) [63]. It measures perceived mental ability where participants provide the frequency of everyday difficulties in the past six months from 0 ‘never’ to 4 ‘very often’. The FLei provides three subscales measuring attention, memory, and executive functioning with 10 items each, and a mental performance sum score consisting of the sum of those scales. The remaining five items focus on visual neglect and will not be taken into account. The FLei scales show high internal consistency (Cronbach’s alphas α ≥ 0.91 [63]). Further, the 75-item Behavior Rating Inventory of Executive Function-Adult Version (BRIEF-A) [64] was used, where participants rated the frequency of behaviors requiring executive functions in the past month from 1 ‘never’ to 3 ‘often’. The BRIEF-A provides a summary score between 70 and 210 (Global Executive Composite, GEC), which consists of two indexes. The first one, the Behavioral-Regulation Index (BRI, score between 30 and 90) measures the ability to regulate behavior and emotions appropriately. The second one, the Metacognition Index (MI, score between 40 and 120) measures the ability to solve problems through planning and organization using active working memory. The Dutch BRIEF-A sum score and indexes show high internal consistency (α ≥ 0.92 [65]). A Dutch adult normative group (*n* = 1600, age 18–65 years) was provided by the questionnaire developers and was used for the computation of percentile scores. Additionally, the BRIEF-A entails three validity scales that indicate, first, unusual or possible noncredible response patterns of negative answers (Negativity), second, the extent to which participants give atypical answers (Infrequency), and third, the extent to which participants give inconsistent answers to similar statements (Inconsistency). 

Depressive symptoms were assessed with the 21-item Beck Depression Inventory (BDI-II) [66]. Participants rate the severity of 21 depressive symptoms in the past 2 weeks with an item score of 0 representing the absence of a symptom or no emotional/behavioral change and an item score of 3 representing a high severity of a symptom or emotional/behavioral change. The Dutch BDI-II provides a sum score, which shows good to excellent internal consistencies in different groups (0.88 ≤ α ≥ 0.92 [67]). According to the manual, individuals scoring between 20 and 28 are considered to show ´moderate´ depressive symptom severity, and individuals scoring 29 or higher to show ´severe´ depressive symptom severity [67].

Sleep was assessed with the 19-item Pittsburgh Sleep Quality Index (PSQI) [68], which measures sleep quality and sleep disturbances in the past month. Participants indicate the time they went to bed and got up in the morning, the number of minutes it took to fall asleep, the hours actually slept, and further rate sleep problems on a scale from 0 to 3 with higher scores indicating more sleep problems. The PSQI provides a global sum score that consists of the seven component scores of sleep quality, sleep latency, sleep duration, habitual sleep efficiency, use of sleeping medication, sleep disturbances, and daytime dysfunction. The PSQI shows good overall internal consistency (α ≥ 0.83); however, the individual coefficients range between 0.35 and 0.76 [68]. The PSQI considers individuals with a global score of ≥5 as bad sleepers. 

The severity of fatigue was assessed with the 9-item Fatigue Severity Scale (FSS) [69], which measures the severity of fatigue symptoms on a 7-point Likert scale from 1 ‘totally disagree’ to 7 ‘completely agree’. The FSS provides one sum score, which shows a good test–retest variability of 0.76 in the Dutch version [70]. The cut-off average score of ≥4 was used as an indication of high fatigue. 

Daily stress was assessed with the Dutch 114-item Alledaagse Problemen Lijst (Everyday Problems List, APLN) [71], which measures the frequency and intensity of daily and chronic stressors in the past two months. Participants state whether they had to deal with a daily stressor (yes or no) and if yes, rate the effect of the stressor on their feelings from 0 ’not bad at all’ to 3 ‘quite bad’. The APLN provides three scales, including a frequency score (FREQ) representing the sum of all chosen stressors, an intensity score (INT) representing the average intensity of all selected stressors, and finally a total sum score (TOT) representing the sum of all chosen items with their intensity. These three scores can be computed from all 114 items, and they can also be computed for a selection of items that refer to events that are dependent on the functioning of the person (DEP, 28 items) and a selection of items that refer to events that are not dependent on the functioning of the person (INDEP, 21 items). This leads to a total of 9 APLN scores. The TOT, FREQ, and INT scores present with good test–retest reliability (0.85, 0.87, and 0.76, respectively) [71]. A Dutch norm group (*n* = 1106) was provided by the questionnaire developers, categorizing percentile scores into ‘very low’, ‘low’, ’normal’, ‘high’, and ‘very high’ categories. 

Coping was assessed with the 32-item COPE-Easy questionnaire [72], which measures strategies to cope with stress and difficulties. Participants rate the applicability of strategies from 1 ‘not applicable to me at all’ to 4 ‘very much applicable to me’. The COPE-Easy consists of three dimensions. The first, active problem-oriented coping (APOC), includes the subscales of active coping, suppression of competing activities, planning, positive reframing, and restraint. The second, support seeking coping (SSC), consists of the subscales of instrumental support, focus on venting emotions, and use of emotional support. The third, avoidance behavior (AB), consists of the subscales of self-distraction, denial, and behavioral disengagement. The four coping strategies of religion, humor, acceptance, and substance use are not allocated to any dimension. The dimensions and single strategies of the COPE-Easy show acceptable to good reliability [72]. Further, five of these strategies can be viewed as facets of problem-focused coping (active coping, suppression of competing activities, planning, restraint, and instrumental support) and another five strategies as aspects of emotion-focused coping (positive reframing, use of emotional support, denial, religion, and acceptance). Three strategies were rated as less useful by the questionnaire developers (venting on emotions, self-distraction, and behavioral disengagement).

### 2.3. Statistical Analysis

All statistical analyses were computed with SPSS version 28. First, raw scores on the questionnaires for the groups are presented with descriptive statistics. To test whether the LFN1, LFN2, and CG show differences in functioning, nonparametric Kruskal–Wallis tests were utilized for overall comparisons and Mann–Whitney U tests were utilized for pairwise comparisons. In order to control for alpha error growth in multiple testing, a strict significance level of *p* < 0.01 commonly used in psychological research was applied. This reduces the chance for a Type I error (leading to a 1% chance of obtaining a false positive result), while limiting the risk for a Type II error. Further, interpretations were more based on effect sizes indicating the magnitude of a finding independent from its statistical significance. Nonparametric tests were chosen since the assumptions of normality and of homogeneity of variance were violated for most variables and groups. Normality was tested through skewness and kurtosis values, the Shapiro–Wilk test, and a visual examination of QQ plots and boxplots. Homogeneity of variance was tested by Levene´s test. The magnitudes of the overall effect of group differences were calculated using the effect size measure eta squared and interpreted as small = 0.01, medium = 0.06, and large = 0.14. [73,74]. The magnitude of pairwise group differences was estimated by Cohen´s r and interpreted as small (0.1 < r < 0.3), medium (0.3 ≤ r < 0.5), or large (r ≥ 0.5) [73]. 

Further, the proportion of individuals with above average symptom reporting on all complaint-related variables (FLei, BRIEF-A, BDI-II, PSQI, FSS, APLN) was computed and compared. Above average symptom reporting was defined as a score above the provided cut-off values proposed by the questionnaire developers for the FSS (≥4) and PSQI (≥5) and when scoring in the categories of ‘moderate’ and ‘severe’ depressive symptom reporting on the BDI-II. For the APLN, above average symptom reporting was considered when scoring in the ‘high’ or ‘very high’ categories provided by the test developers, corresponding to a percentile score of 80 or higher. For the remaining questionnaires (FLei and BRIEF-A), above average symptom reporting was defined as a score one standard deviation above the mean, thus a score equal to or above the 84th percentile. The BRIEF-A provides a normative group with percentile scores, from which the percentile cut-off of 84% could be directly applied. The FLei does not provide normative data and the healthy control sample used by the questionnaire developers was markedly smaller than our comparison sample (*n* = 97), as well as younger (42.5 years), and presented with somewhat less females (59%) [63]. Accordingly, a cut-off score of one standard deviation above the mean scores of the CG of this present study was used [63]. Finally, summarized domain scores were computed for the cognition, depressive symptoms, sleep, fatigue, and stress domains, showing the proportion of individuals with above average scores on at least one of the domain-specific variables. The frequency of individuals with above average symptom reporting in all groups was compared with Chi-Square tests at a significance level of *p* < 0.01 to control for alpha error growth in multiple testing. The size of associations between variables was computed using Cramer’s V and interpreted as small (0.1), medium (0.3), and large (0.5) for 1 degree of freedom and as small (0.07), medium (0.21), and large (0.35) for 2 degrees of freedom [73]. Finally, the number of functional domains for which participants would show above average symptom reporting was calculated based on those summarized domain scores.

## 3. Results

For the raw scores, descriptive statistics, group comparisons, and effect sizes on all complaint variables are presented in Table 2. The proportion of individuals with above average symptom reporting, group differences, and effect sizes for all complaint variables are presented in Table 3. Further, Figure 1 depicts the proportion of individuals in each group showing above average symptom reporting on the five functional domains. Finally, descriptive statistics of the coping strategies, group comparisons, and effect sizes are depicted in Table 4. In the following sections, the overall terms LFN2 and CG are used for different subgroups based on the outcome variables. That means that the term LFN2 and CG will refer to the subgroups LFN2-SA and CG-SA for the measures of cognitive functioning and depressive symptoms, and the subgroups LFN2-SB and CG-SB for the measures of sleep, fatigue, stress, and coping.

The proportion of missing values differed between the three groups and between the single questionnaires (see Appendix A). With the online forced entry fill-out format, the LFN2 group and CG showed no missing values, except for on the PSQI sleep questionnaire. Proportions of missing values on the PSQI subcomponents ranged from 0 to 5%. No global score could be computed for 6% of the LFN2 group and 3% of the CG. The LFN1 group showed missing values on all questionnaires. The highest proportions were observed on the PSQI subcomponents (0–11%) and the PSQI global score (22%). This high number of missing values on the PSQI is, to a great extent, due to the four open questions regarding the time that participants spent in bed and spent sleeping. If not filled out in the correct format, some subcomponent scores, and consequently the global score, could not be computed. Proportions of missing values on the other questionnaires ranged between 0 and 9%.

In total, 50 participants (9.5%) scored above the cut-offs for invalid answering patterns on the three BRIEF-A validity scales, suggesting possible noncredible reports. An overview per group and scale is provided in Appendix A. Since only some groups completed the BRIEF-A (LFN1, LFN2-SA, and CG-SA), and the proportion of individuals with possibly noncredible answer patterns in the LFN2-SB and CG-SB could therefore not be deduced, these 50 individuals were retained in the data set. In order to control for the effect of including those participants in the study, all subsequently described results were also computed with a data set excluding those participants (please refer to Appendix A). No or only very small changes were observed that had no meaningful effect on the outcomes. 

### 3.1. LFN-Related Cognitive Complaints

#### 3.1.1. Raw Scores

The LFN1 group reported the most complaints on all variables, followed by the LFN2 group and the CG. Specifically, the three groups differed overall significantly on all cognitive variables with large effect sizes (Table 2). Only the BRIEF-A Behavioral Regulation showed a medium effect size. Furthermore, all pairwise comparisons were significant. These comparisons reached between the LFN1 group and CG large (FLei Sum and attention score) and medium (all other comparisons) effect sizes. Between the LFN2 group and CG, medium effect sizes were seen on the FLei sum, attention, and memory scales.

#### 3.1.2. Proportion of Individuals with above Average Symptom Reporting

The LFN1 group showed the highest proportions of above average symptom reporting (23 to 64%), followed by the LFN2 group (15 to 44%) and the CG (5 to 18%) (Table 3). All overall differences were significant with large (FLei sum, attention, and memory) and medium effect sizes (all executive function scales). Notable significant pairwise differences of a medium effect size were found between the LFN1 group and CG on all FLei scales and between the LFN2 group and CG on the FLei memory scale. Finally, the groups differed significantly on the summarized cognition domain score with a large effect size. Further, a significant pairwise comparison of a medium effect size between the LFN1 group and CG was observed (Table 3, and a visual overview in Figure 1).

**Table 2 jcm-13-00935-t002:** Descriptive characteristics, significance tests, and effect sizes on all outcome variables between the three groups.

	LFN1 *n* = 181				LFN2 LFN2-SA = 131				CG CG-SA = 229							LFN1–LFN2	LFN1–CG	LFN2–CG
	N	M ± SD	Range	M	N	M ± SD	Range	M	N	M ± SD	Range	M	*H*	*p*	*η^2^*	*r*	*r*	*r*
**Cognition**																		
FLei Sum	169	43.2 ± 23.2	0–108	42	131	31.3 ± 19.1	0–81	30	229	18.9 ± 14.9	0–65	17	118.4	<0.001 **	0.22	0.25 **	0.52 **	0.32 **
FLei Attention	170	15.6 ± 8.8	0–36	16	131	10.6± 6.9	0–32	10	229	6.0 ± 5.3	0–24	5	131.5	<0.001 **	0.25	0.28 **	0.53 **	0.34 **
FLei Memory	177	15.5 ± 7.9	0–35	15	131	11.8 ± 6.5	0–28	11	229	7.7 ± 5.9	0–27	7	106.7	<0.001 **	0.20	0.23 **	0.49 **	0.31 **
Executive functions																		
FLei Executive functions	180	11.7 ± 7.7	0–37	11	131	8.9 ± 6.5	0–25	8	229	5.2 ± 4.7	0–20	4	86.6	<0.001 **	0.16	0.17 *	0.45 **	0.28 **
BRIEF-A Global	167	109.9 ± 23.1	70–168	107	131	99.9 ± 22.7	70–162	98	229	89.7 ± 17.0	70–148	86	80.8	<0.001 **	0.15	0.22 **	0.45 **	0.22 **
BRIEF-A BR	172	46.6 ± 10.3	30–77	46	131	42.8 ± 9.6	30–67	41	229	38.5 ± 7.7	30–67	37	70.0	<0.001 **	0.13	0.19 **	0.41 **	0.23 **
BRIEF-A MC	172	63.0 ± 14.2	40–104	61	131	57.1 ± 14.1	40–95	55	229	51.2± 10.6	40–86	48	76.7	<0.001 **	0.14	0.22 **	0.44 **	0.20 **
**Depressive symptoms**																		
BDI-II Sum	178	11.7 ± 8.3	0–42	10	131	7.2 ± 6.2	0–29	6	229	4.4 ± 4.3	0–27	3	104.7	<0.001 **	0.19	0.29 **	0.50 **	0.24 **
					**LFN2-SB = 108**					**CG-SB = 239**								
**Sleep (PSQI)**																		
Global	142	8.6 ± 4.7	1–19	8	102	7.7 ± 4.0	0–19	7	231	5.5 ± 3.3	0–19	5	49.0	<0.001 **	0.10	0.08	0.33 **	0.27 **
Sleep Quality	178	1.6 ± 0.9	0–3	2	108	1.3 ± 0.8	0–3	1	239	0.9 ± 0.8	0–3	1	54.0	<0.001 **	0.10	0.17 *	0.35 **	0.19 **
Sleep Latency	164	1.5 ± 1.1	0–3	1	103	1.5 ± 1.1	0–3	1	232	1.0 ± 1.0	0–3	1	29.1	<0.001 **	0.05	0.01	0.24 **	0.22 **
Sleep Duration	170	1.1 ± 1.1	0–3	2	106	0.9 ± 1.0	0–3	1	239	0.5 ± 0.8	0–3	0	32.2	<0.001 **	0.06	0.08	0.27 **	0.19 **
Habit Sleep Efficiency	169	1.5 ± 1.2	0–3	1	105	1.5 ± 1.3	0–3	1	236	1.1 ± 1.2	0–3	1	10.2	0.006 *	0.01	0.02	0.15 *	0.11
Sleep Disturbance	161	1.4 ± 0.6	0–3	1	108	1.4 ± 0.5	0–3	1	239	1.2 ± 0.5	0–3	1	19.3	<0.001 **	0.03	0.06	0.21 **	0.15 *
Sleep Medication	181	0.8 ± 1.2	0–3	0	108	0.4 ± 0.9	0–3	0	239	0.3 ± 0.8	0–3	0	34.1	<0.001 **	0.06	0.19 *	0.28 **	0.08
Daytime Dysfunction	178	1.0 ± 0.8	0–3	1	108	0.8 ± 0.7	0–3	1	239	0.5 ± 0.6	0–3	0	39.1	<0.001 **	0.07	0.06	0.28 **	0.23 **
**Fatigue**																		
FSS Sum	180	37.0 ± 13.9	9–63	38	108	36.0 ± 10.7	13–62	37	239	27.1 ± 11.6	9–61	25	68.4	<0.001 **	0.12	0.05	0.35 **	0.35 **
**Daily stress (APLN)**																		
All items																		
Total	165	38.1 ± 32.2	0–172	27	108	41.7 ± 42.1	0–198	27	239	18.6 ± 35.7	0–248	8	110.0	<0.001 **	0.21	0.01	0.48 **	0.40 **
Frequency	169	26.5 ± 17.9	0–114	22	108	45.4 ± 31.5	1–114	37	239	22.2 ± 22.4	0–114	16	71.8	<0.001 **	0.14	0.32 **	0.20 **	0.43 **
Intensity	164	1.4 ± 0.6	0–3.0	1.3	108	0.9 ± 0.5	0.0–2.3	0.8	229	0.7 ± 0.5	0.0–3.0	0.6	141.6	<0.001 **	0.28	0.44 **	0.58 **	0.21 **
Dependent items																		
Total	171	8.5 ± 8.6	0–37	6	108	9.4 ± 10.6	0–50	5	239	4.1 ± 8.6	0–63	2	77.7	<0.001 **	0.15	0.01	0.38 **	0.37 **
Frequency	175	6.1 ± 5.1	0–28	5	108	10.9 ± 8.1	0–28	9	239	5.5 ± 5.7	0–28	4	53.2	<0.001 **	0.10	0.32 **	0.10	0.39 **
Intensity	156	1.3 ± 0.7	0.0–3.0	1.3	106	0.8 ± 0.5	0.0–2.0	0.7	215	0.6 ± 0.6	0.0–3.0	0.4	118.6	<0.001 **	0.25	0.42 **	0.54 **	0.21 **
Independent items																		
Total	172	8.4 ± 6.6	0–37	7	108	8.5 ± 8.4	0–40	6	239	4.0 ± 7.5	0–52	2	107.2	<0.001 **	0.20	0.06	0.48 **	0.37 **
Frequency	174	5.3 ± 3.6	0–21	5	108	8.4 ± 5.9	0–21	6.5	239	4.3 ± 4.3	0–21	3	60.1	<0.001 **	0.11	0.26 **	0.21 **	0.39 **
Intensity	167	1.6 ± 0.7	0–3.0	1.6	106	1.0 ± 0.6	0.0–2.7	1.0	211	0.7 ± 0.6	0.0–3.0	0.7	125.6	<0.001 **	0.26	0.44 **	0.55 **	0.18 *

Note: LFN1 = LFN group recruited via LFN foundation, LFN2 = LFN group recruited via online panel, SA = subsample A filling out questionnaires regarding cognition and depressive symptoms, SB = subsample B filling out questionnaires regarding sleep, fatigue, stress, and coping, CG = Comparison group recruited via online panel, FLei = Questionnaire for Complaints of Cognitive Disturbances, BRIEF-A = Behavior Rating Inventory of Executive Function—Adult Version, BRIEF-A BR = BRIEF-A Behavioral Regulation Index, BRIEF-A MC = BRIEF-A Metacognition, BDI-II = Beck Depression Inventory, PSQI = Pittsburgh Sleep Quality Index, FSS = Fatigue Severity Scale, APLN = Alledaagse Problemen Lijst, H = Kruskal–Wallis statistic for testing overall group differences, η^2^ = eta squared, r—Effect size Cohen’s r shown with the significance level derived from pairwise Mann–Whitney U tests based on: * significant difference at a level *p* < 0.01, ** significant difference at a level *p* < 0.001, A positive r value was used when the firstly mentioned group had more complaints,    = medium effect size,    = large effect size.

**Table 3 jcm-13-00935-t003:** Proportions and number of individuals with above average symptom reporting, significance tests, and effect sizes on all outcome variables between the three groups.

	LFN1 *n* = 181		LFN2 LFN2-SA = 131		CG CG-SA = 229						LFN1–LFN2	LFN1–CG	LFN2–CG
	N	%	N	%	N	%	*χ2*	*df*	*p*	*V*	*V*	*V*	*V*
**Cognition ^a^**	**181**	**75.7**	**131**	**61.1**	**229**	**33.2**	**76.88**	**2**	**<0.001 ****	**0.38**	**0.16 ***	**0.42 ****	**0.27 ****
FLei Sum	169	62.1	131	44.3	229	17.9	82.68	2	<0.001 **	0.40	0.18 *	0.45 **	0.28 **
FLei Attention	170	64.1	131	39.7	229	16.2	96.31	2	<0.001 **	0.43	0.24 **	0.49 **	0.26 **
FLei Memory	177	58.8	131	41.2	229	14.8	86.05	2	<0.001 **	0.40	0.17 *	0.46 **	0.30 **
Executive functions													
FLei Executive functions	180	52.2	131	43.3	229	17.9	56.58	2	<0.001 **	0.32	0.09	0.36 **	0.28 **
BRIEF-A Global	167	23.2	131	15.3	229	5.2	31.93	2	<0.001 **	0.25	0.12	0.29 **	0.17 *
BRIEF-A BR	172	23.8	131	14.5	229	6.1	25.82	2	<0.001 **	0.22	0.12	0.26 **	0.14 *
BRIEF-A MC	172	28.5	131	17.6	229	7.9	29.79	2	<0.001 **	0.24	0.13	0.27 **	0.15 *
**Depressive symptoms (BDI-II)**	**178**	**18.5**	**131**	**5.3**	**229**	**0.9**	**44.89**	**2**	**<0.001 ****	**0.29**	**0.19 ****	**0.31 ****	**0.14 ***
Moderate symptoms	178	14.6	131	4.6	229	0.9	32.79	2	<0.001 **	0.25	0.16 *	0.27 **	0.12
Severe symptoms	178	3.9	131	0.8	229	0	11.19	2	0.004 *	0.14	0.10	0.15 *	0.07
			**LFN2-SB = 108**		**CG-SB = 239**								
**Sleep (PSQI)**	142	**75.4**	**102**	**79.4**	**231**	**55.0**	**26.31**	**2**	**<0.001 ****	**0.24**	**0.05**	**0.21 ****	**0.23 ****
**Fatigue (FSS)**	180	**56.7**	**108**	**58.3**	**239**	**26.4**	**50.99**	**2**	**<0.001 ****	**0.31**	**0.02**	**0.31 ****	**0.31 ****
**Daily stress (APLN)**	**169**	**61.5**	**108**	**77.8**	**239**	**29.7**	**81.70**	**2**	**<0.001 ****	**0.40**	**0.17 ***	**0.32 ****	**0.45 ****
Total	165	41.8	108	41.7	239	11.7	57.39	2	<0.001 **	0.34	0.002	0.35 **	0.34 **
Frequency	169	46.2	108	75.0	239	28.9	64.58	2	<0.001 **	0.35	0.29 **	0.18 **	0.43 **
Intensity	164	34.1	108	6.5	229	3.1	83.27	2	<0.001 **	0.41	0.32 **	0.42 **	0.08

Note: ^a^ Overall categories in bold letters refer to participants who are categorized as reporting above average symptom reporting on at least one of the underlying variables. LFN1 = LFN group recruited via LFN foundation, LFN2 = LFN group recruited via online panel, SA = subsample A filling out questionnaires regarding cognition and depressive symptoms, SB = subsample B filling out questionnaires regarding sleep, fatigue, stress, coping, CG = Comparison group recruited via online panel, FLei = Questionnaire for Complaints of Cognitive Disturbances, BRIEF-A = Behavior Rating Inventory of Executive Function—Adult Version, BRIEF-A BR = BRIEF-A Behavioral Regulation Index, BRIEF-A MC = BRIEF-A Metacognition, BDI-II = Beck Depression Inventory, PSQI = Pittsburgh Sleep Quality Index, FSS = Fatigue Severity Scale, APLN = Alledaagse Problemen Lijst, % = Percentages from the total of valid cases, *V*—Effect size Cramer’s V shown with the significance level of the group comparison based on: * significant difference at a level *p* < 0.01, ** significant difference at a level *p* < 0.001,    = medium effect size,    = large effect size.

**Figure 1 jcm-13-00935-f001:**
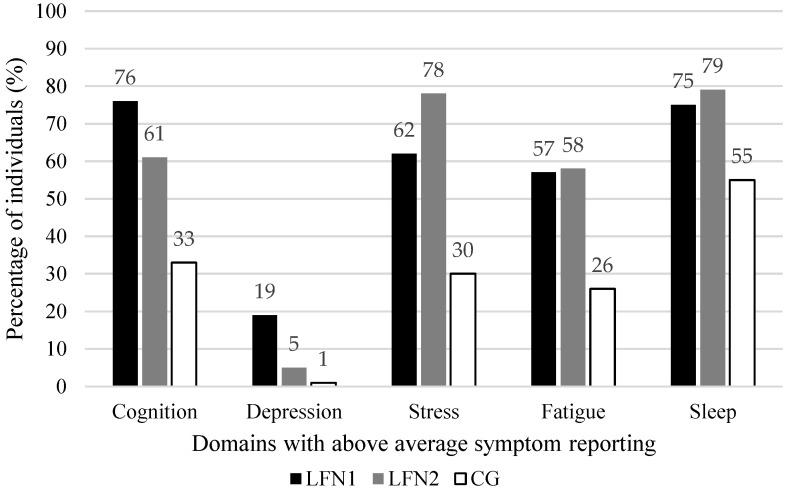
Proportion of individuals that show above average symptom reporting on each of the five functional domains in the three groups.

### 3.2. LFN-Related Psychological Complaints

#### 3.2.1. Depressive Symptoms

##### Raw Scores

In terms of depressive symptoms, the LFN1 group reported the most depressive symptoms, followed by the LFN2 group and the CG (Table 2). The groups differed overall significantly with a large effect size. Also, all pairwise comparisons were significant, with one medium effect size between the LFN1 group and CG.

##### Proportion of Individuals with above Average Symptom Reporting

A moderate depressive symptomatology was observed for 15% of the LFN1 group, 5% of the LFN2 group, and 1% of the CG (Table 3). The overall group difference was significant, with a medium effect size. Pairwise comparisons were also significant, but with small effect sizes. A severe depressive symptomatology was observed for 4% of the LFN1 group, 1% of the LFN2 group, and 0% of the CG. The only significant difference was observed for the pairwise comparison between the LFN1 group and CG, however, with a small effect size. On the summarized depressive symptom domain score, the groups differed significantly overall with a medium effect size (Table 3, Figure 1). Pairwise comparisons were significant, but only reached a medium effect size between the LFN1 group and CG.

#### 3.2.2. Sleep

##### Raw Scores

The LFN1 group reported the most sleep difficulties, followed by the LFN2 group and CG (Table 2). However, the LFN1 and LFN2 groups scored similarly on sleep latency, habitual sleep efficiency, and sleep disturbance. Overall, the three groups differed significantly on all sleep variables, but mostly with small effect sizes. Medium effect sizes were only observed for sleep quality, daytime dysfunction, and the global score. Further pairwise comparisons showed mostly nonsignificant differences between the LFN1 and LFN2 groups. Significant pairwise differences between the LFN groups and the CG were observed, however with small effect sizes. Only the global score and sleep quality score reached a medium effect size between the LFN1 group and CG. 

##### Proportion of Individuals with above Average Symptom Reporting

In terms of the proportion of individuals with high reported sleeping difficulties (Table 3, Figure 1), the highest proportion of bad sleepers was observed in the two LFN groups (LFN1 = 75%, LFN2 = 79%), followed by the CG (55%). The overall difference was significant with a medium effect size. The pairwise comparisons showed a significant difference between the two LFN groups with the CG, but with small effect sizes.

#### 3.2.3. Fatigue

##### Raw Scores

Both LFN groups scored significantly higher on fatigue symptom severity compared to the CG. Overall, the groups differed significantly with a medium effect size (Table 2). Pairwise comparisons reached medium effect sizes between the LFN groups and the CG; the LFN1 and LFN2 groups did not differ significantly from each other. 

##### Proportion of Individuals with above Average Symptom Reporting

Twice as many individuals in the two LFN groups reported high levels of fatigue (LFN1 = 57%, LFN2 = 58%) compared to the CG (26%) (Table 3, see also Figure 1). The overall group difference, as well as the pairwise comparisons between the two LFN groups and the CG, were significant and of a medium effect size. 

#### 3.2.4. Daily Stress

##### Raw Scores

On the intensity scale, the LFN1 group reported the highest stress intensity, followed by the LFN2 group and CG (Table 2). The three groups differed significantly overall, with a large effect size. Pairwise comparisons were significant and of a large effect size between the LFN1 group and CG, and a medium effect size between the LFN1 and LFN2 groups. A different pattern was observed for the frequency scale where the LFN2 group scored the highest compared to the other two groups. The three groups differed overall significantly with a medium effect size. Pairwise comparisons were significant and reached a medium effect size between the LFN1 and LFN2 groups, and the LFN2 group and CG. Yet another pattern was observed on the total score, where the two LFN groups scored similarly, but both were significantly higher than the CG. The overall group difference was significant with a large effect size. Pairwise comparisons were significant with medium effect sizes between the LFN1 group and the CG, and the LFN2 group and CG. The results that consider only dependent or only independent items show the same result patterns as the results that consider all items.

##### Proportion of Individuals with above Average Symptom Reporting

Considering the number of individuals with above average reported stress symptoms, a similar pattern can be observed (Table 3). The LFN1 group presents with considerably more individuals with above average stress intensity (34%) compared to the other groups (LFN2 = 7%, CG = 3%). This overall group difference reached a large effect size and pairwise comparisons between the LFN1 group with the other groups reached medium effect sizes. On the frequency scale, the LFN2 group presented with the most individuals with a high frequency of stressors (75%) compared to the LFN1 group (46%) and the CG (29%). The overall group comparison reached a large effect size and the pairwise comparison between the LFN2 group and CG reached a medium effect size. On the total scale, the two LFN groups showed similarly higher proportions of high stress reporting (LFN1 = 42%, LFN2 = 42%) compared to the CG (12%). The overall group difference, as well as the pairwise group differences between the two LFN groups with the CG, reached medium effect sizes. Finally, on the summarized daily stress domain score, the groups differed overall significantly with a large effect size, and with pairwise comparisons of a medium effect size between the two LFN groups and the CG (Table 3, Figure 1).

### 3.3. Sum of Domains with above Average Symptom Reporting

Finally, the sum of all functional domains for which an individual shows above average symptom reporting was computed. From the LFN1 group, 1.5% would not show above average reporting in any of the domains. About 15% would show above average reporting in one domain, 45% in two domains, and 39% in three or more domains. Notably, for 27% of the LFN1 group, no sum score could be computed due to missing values on at least one of the domains. Due to the splitting of the questionnaire, no sum score across all domains could be computed for the LFN2 group and CG. When only considering the first half of the questionnaires (cognition and depressive symptoms), 39% of the LFN2 group showed no above average symptom reporting on any of the domains, followed by 56% showing above average symptom reporting in one domain, and 5% in two domains. The proportions in the CG were 67% for no above average symptoms on any domain, 32% in one domain, and 1% in two domains. When considering the second half of the questionnaires (sleep, fatigue, and stress), 11% of the LFN2 group showed no above average symptom reporting on any of the domains, followed by 45% showing above average symptom reporting in one domain, and 44% in two or more domains. In the CG, 13% showed no above average symptoms in any domains, 36% in one domain, and 52% in two or more domains.

### 3.4. Coping Strategies

When comparing the strategies chosen as most applicable, we observe similarities, but also some differences between the groups (Table 4). As a general trend, the LFN1 group reported using the different coping strategies the most, followed by the LFN2 group and the CG. 

**Table 4 jcm-13-00935-t004:** Descriptive characteristics, significance tests, and effect sizes on the Cope-Easy questionnaire between the three groups.

	LFN1 *n* = 181				LFN2 LFN2-SB = 108				CG CG-SB = 239							LFN1–LFN2	LFN1–CG	LFN2–CG
	N	M ± SD	Range	M	N	M ± SD	Range	M	N	M ± SD	Range	M	*H*	*p*	*η^2^*	*r*	*r*	*r*
**Active problem-oriented coping ^a^**	**169**	**24.6 ± 6.5**	**10–39**	**24**	**108**	**21.8 ± 7.2**	**10–40**	**21**	**239**	**21.0 ± 7.3**	**10–40**	**21**	**24.7**	**<0.001 ****	**0.04**	**0.21 ****	**0.24 ****	**0.04**
Active coping ^b^	179	6.0 ± 1.9	2–8	6	108	5.0 ± 1.9	2–8	5	239	4.8 ± 2.0	2–8	5	36.7	<0.001 **	0.07	0.25 **	0.29 **	0.04
Suppression of competing activities ^b^	179	4.2 ± 1.8	2–8	4	108	4.0 ± 1.8	2–8	4	239	3.9 ± 1.8	2–8	4	2.7	0.26	<0.01	0.05	0.08	0.03
Positive reframing ^c^	178	4.7 ± 2.0	2–8	4	108	4.3 ± 1.8	2–8	4	239	4.3 ± 1.9	2–8	4	4.5	0.10	<0.01	0.09	0.10	0.02
Planning ^b^	178	5.8 ± 1.9	2–8	6	108	4.5 ± 1.8	2–8	4	239	4.5 ± 1.9	2–8	4	53.8	<0.001 **	0.10	0.33 **	0.33 **	<0.01
Restrain ^b^	169	4.0 ± 1.6	2–8	4	108	4.0 ± 1.6	2–8	4	239	3.6 ± 1.6	2–8	3	9.0	0.01	0.01	<0.01	0.13 *	0.12
**Support Seeking**	**178**	**15.0 ± 4.4**	**6–24**	**15**	**108**	**11.0 ± 4.0**	**6–22**	**10**	**239**	**10.0 ± 4.0**	**6–24**	**9**	**122.6**	**<0.001 ****	**0.23**	**0.43 ****	**0.52 ****	**0.14 ***
Instrumental support ^b^	179	4.7 ± 1.8	2–8	5	108	3.4 ± 1.6	2–8	3	239	3.0 ± 1.5	2–8	2	95.1	<0.001 **	0.18	0.34 **	0.47 **	0.14
Focus on venting emotions ^d^	178	4.9 ± 1.8	2–8	5	108	3.7 ± 1.4	2–8	3	239	3.4 ± 1.5	2–8	3	84.1	<0.001 **	0.16	0.35 **	0.43 **	0.11
Use of emotional support ^c^	178	5.4 ± 1.9	2–8	6	108	3.8 ± 1.6	2–8	4	239	3.5 ± 1.7	2–8	3	99.4	<0.001 **	0.19	0.39 **	0.47 **	0.11
**Avoidance Behavior**	**176**	**10.7 ± 3.2**	**6–21**	**10**	**108**	**10.3 ± 3.3**	**6–23**	**10**	**239**	**9.1 ± 2.8**	**6–19**	**9**	**30.8**	**<0.001 ****	**0.06**	**0.08**	**0.27 ****	**0.17 ***
Self-distraction ^d^	178	4.7 ± 1.8	2–8	5	108	4.2 ± 1.8	2–8	4	239	3.6 ± 1.6	2–8	3	36.5	<0.001 **	0.07	0.14	0.29 **	0.14 *
Behavioral disengagement ^d^	178	3.1 ± 1.5	2–8	2	108	3.0 ± 1.4	2–8	2	239	2.6 ± 1.0	2–8	2	9.5	0.009 *	0.01	0.02	0.14 *	0.11
Denial ^c^	178	3.0 ± 1.5	2–8	2	108	3.1 ± 1.6	2–8	2	239	2.8 ± 1.5	2–8	2	7.0	0.03	0.01	0.06	0.09	0.14
Non-dimension bound strategies																		
Religion ^c^	179	2.8 ± 1.7	2–8	2	108	3.0 ± 1.6	2–8	2	239	2.6 ± 1.5	2–8	2	7.7	0.02	0.01	0.10	0.05	0.15 *
Humor	177	3.9 ± 1.7	2–8	4	108	4.0 ± 1.7	2–8	4	239	3.7 ± 1.7	2–8	4	1.8	0.40	<0.01	0.03	0.04	0.07
Acceptance ^c^	178	4.7 ± 1.8	2–8	4.5	108	4.9 ± 1.6	2–8	5	239	4.8 ± 2.0	2–8	5	1.2	0.54	<0.01	0.07	0.01	0.05
Substance use	178	4.9 ± 2.1	4–16	4	108	4.9 ± 1.5	4–11	4	239	4.3 ± 0.7	4–9	4	24.2	<0.001 **	0.04	0.05	0.19 **	0.24 **

Note: ^a^ Dimension scores are depicted in bold letters encompassing the underlying subscales, ^b^ Problem-focused coping strategies, ^c^ Emotion-focused coping strategies, ^d^ Less useful strategies as rated by the test developers, LFN1 = LFN group recruited via LFN foundation, LFN2 = LFN group recruited via online panel, SB = subsample B filling out questionnaires regarding stress, coping, fatigue, and sleep, CG = Comparison group recruited via online panel, H = Kruskal–Wallis statistic for testing overall group differences, η^2^ = eta squared, r—Effect size Cohen’s r shown with the significance level derived from pairwise Mann–Whitney U tests based on: * significant difference at a level *p* < 0.01, ** significant difference at a level *p* < 0.001, In this table, a positive r value was used when the firstly mentioned group rated coping strategies as more applicable to them,    = medium effect size,    = large effect size.

#### 3.4.1. Active Problem-Oriented Coping

The LFN1 group reported using significantly more active problem-oriented coping strategies compared to the other two groups. However, differences were of a small effect size. When looking at the subscales of active problem-oriented coping, the three groups differed overall significantly with a medium effect size in terms of active coping and planning. On the pairwise comparisons, the LFN1 group scored significantly higher on active coping and planning compared to the other two groups. However, these differences were of a medium effect size only for planning. No other pairwise significant differences were observed. 

#### 3.4.2. Support Seeking

The three groups differed overall with large effect sizes on the dimension and all its subscales. On the pairwise comparisons, the LFN1 group reported significantly higher scores on all support seeking scales compared to the other two groups with medium effect sizes. A large effect size was observed only on the dimension score between the LFN1 group and CG. 

#### 3.4.3. Avoidance Behavior

Although the LFN1 group reported using the avoidance strategies the most, followed by the LFN2 group, differences were small. The overall group differences on the dimension score and on the self-distraction and behavioral disengagement subscales reached significance; however, only self-distraction reached a medium effect size. None of the pairwise comparisons reached medium effect sizes. 

#### 3.4.4. Coping Strategies Not Allocated to Any Dimension

Considering the four single coping domains, the only significant difference was seen on substance use. The two LFN groups used significantly more substances compared to the CG; however, none of these differences reached a medium effect size. 

## 4. Discussion

Various subjective complaints with possibly high daily life burden have been reported in association with LFN. However, previous studies do not allow for firm conclusions about the frequency and extent of such complaints and some of these complaints have not yet been thoroughly investigated with structured, standardized, and psychometrically validated measurement instruments. In this present study, the subjectively reported complaints of cognitive difficulties, depressive symptoms, sleeping difficulties, fatigue, and daily stress of people reporting LFN perceptions compared to those not reporting LFN perceptions were assessed. Further, this study aimed at a multi-faceted investigation of coping strategies used in daily life. 

### 4.1. LFN-Related Cognitive Complaints

In terms of cognition, the most difficulties were reported by the LFN1 group on all cognitive domains, although the LFN2 group also reported more difficulties than the CG. The largest group differences and the highest proportion of individuals with above average symptom reporting were observed on overall cognitive functioning, followed by the specific domains of attention, and then memory difficulties. The lowest symptom reporting was observed in terms of executive functions. The findings about attention difficulties are in line with previous survey studies reporting concentration difficulties as a common complaint in relation to LFN [5,6,8,11,13]. However, the second highest reported cognitive difficulties were memory problems, which have not been commonly included in previous research or regarded as a frequent complaint [21]. In a previous publication based on the larger LFN research project that this current study is part of, only 2.6% of the participants in an overlapping sample to the LFN1 group indicated memory complaints when answering an open question about experienced psychological problems [62]. Thus, the results of this research indicate that, when measured with standardized instruments, subjective memory difficulties are more common in individuals with reported daily LFN perceptions than formerly assumed, and more attention should be paid to this complaint in the future. Furthermore, this research was the first, to our knowledge, to investigate subjective executive function complaints. Higher subjective complaints and higher proportions of participants with above average symptom reporting in the LFN groups suggest that further attention has to be paid to these aspects of cognition as well. However, executive functions seem to be the least common cognitive complaint and memory and especially attention difficulties seem to be more prevalent. Interestingly, the present research found that cognition was, besides sleeping difficulties, the complaint where most participants in the LFN1 group showed above average symptom severity (76%). Subsequently, cognitive complaints related to experiencing LFN should be of importance for future research, as well as for environmental and healthcare services. However, previous research using objective cognitive measures presents different findings on all cognitive domains [29] and further research is necessary to observe to what extent these subjective reports can be associated with objective cognitive measures. Also, LFN exposure was suggested to negatively impact higher-order cognitive functions specifically [28]. Accordingly, future research on the complexity of affected cognitive functions would be relevant.

### 4.2. LFN-Related Psychological Complaints

#### 4.2.1. Depressive Symptoms

The highest symptom severity and highest proportions of moderate or severe symptomatology were reported by the LFN1 group. This was followed by the LFN2 group and then the CG, with both reporting considerably fewer depressive symptoms than the LFN1 group. Correspondingly, the largest group differences were found between the LFN1 and CG. Such group differences were also found by Mirowska and Mroz [31], who used the same depressive symptom questionnaire. However, the proportions of moderate and severe symptomatology combined were markedly higher in their groups (LFN = 30%, CG = 5%) compared to all of our groups (LFN1 = 18%, LFN2 = 5%, CG = 1%). While no one in the CG of both studies showed severe depressive symptoms, 11% of their LFN complainants showed severe symptom severity compared to only 4% in our LFN1 and 1% in our LFN2 group. A possible explanation for this discrepancy could lie in the sampling method, since Mirowska and Mroz conducted acoustical measurements and determined a noise source for their participants. Our current study was based on subjective reports of LFN and probably included a more heterogeneous LFN complainant group with a wider range of symptoms and symptom severity. Furthermore, it has to be noted that the low percentages of depressive symptomatology in the LFN2 group and CG are also explainable by our exclusion criteria of not having any diagnosis of a depressive disorder. Considering this, it is striking that the LFN2 group still presented with significantly more depressive symptoms than the CG.

The current estimated proportion of Dutch adults that suffer from a depressive disorder at any time during a year (year prevalence 2019–2022) lies at 8.5% [75]. From that, it could be assumed that our LFN1 group could present with a higher proportion of depressive disorders compared to the general population. Thus, it could be hypothesized that a high depressive symptomatology is becoming especially prevalent in individuals with higher general hindrances. However, it has to be considered that a higher depressive symptom severity, as measured in this research, is not to be equated with a diagnosis of depression by a clinician. The proportions of self-reported depressive feelings in relation to LFN ranged from 19% [35] to 53% [32] in previous research and 46% of the overlapping sample to the LFN1 group in Erdélyi et al. [62] reported LFN-related depressed mood on a multiple-choice question. In contrast, a diagnosis of a depressive disorder by a clinician was only reported by 7% of the latter sample [62]. Notably, this latter proportion is still lower than the Dutch adult prevalence. Further, the prevalence for depressive disorders is higher for females (10.4%) than males (6.7%) in the general population [7], which aligns with the fact that LFN complainants in our study and other (survey) studies seem to be presenting with more females (see also [5,6,8]). 

Finally, our results are not in accordance with some previous studies [33,34,35] which did not find associations between (assumed) noise exposure and both reported symptoms of depression or a diagnosis of depression. This could indicate that a depressive symptomatology might be less predictable by LFN exposure itself. Rather, a depressive symptomatology could be a secondary symptom, which also depends on other non-acoustic factors that have been shown by noise research to play a highly relevant role in predicting noise-related health outcomes, such as noise annoyance [5,56,57].

Overall, a high depressive symptomology was the least frequently observed complaint in both LFN groups. However, it is among the complaints with a very high possible impact on the quality of life of affected individuals and their surroundings [30]. More attention towards and research into depressive symptomology, a diagnosis of depression by a clinician, and the role of possible moderators and mediators are necessary.

#### 4.2.2. Sleep

The LFN1 group reported the most complaints on all sleep variables, followed by the LFN2 group, who reported less, or on some variables, the same extent of difficulties. The CG showed the least complaints with regard to sleep. These findings are in line with the large body of previous research considering sleeping difficulties as one of the main complaints in relation to LFN (e.g., [6,8,13,31]). The most problems reported by the two LFN groups were in terms of sleep quality, sleep latency (i.e., the time to fall asleep), habitual sleep efficiency (i.e., the time spent asleep compared to the time spent in bed), and sleep disturbance. Less prevalent difficulties were observed in terms of sleep duration, sleep medication usage, and daytime dysfunction. This differential effect of LFN on specific aspects of sleep also aligns with previous findings. Considering sleep quality, worse sleep quality during nocturnal LFN exposure has been suggested in the study by Persson-Waye and colleagues [38], worse subjective sleep quality was reported (despite better objectively measured sleep quality) in the study by Öhrström and colleagues [40], and Ising and Ising related altered cortisol levels in their study to lower sleep quality [14]. Further, difficulties with sleep latency and sleep disturbances were observed in previous research by Persson-Waye and colleagues [13,38], although another study by them did not find those aspects of sleep to be affected [39]. Interestingly, daytime dysfunction due to sleepiness was among the least reported problems in all groups in the present study. However, groups differed significantly and with a medium effect size. To our knowledge, long-term sleep-related daytime dysfunction in the daily life of individuals with LFN perceptions has not yet been specifically investigated. A possible indication for reduced daytime dysfunction could be seen in the lower-rated degrees of daytime activity that was associated with attenuated cortisol levels 30 min post-awakening after nocturnal LFN exposure [38]. Also, the associations of nocturnal LFN exposure with negative mood [38,39], morning tiredness [38,39], and morning tension [13] (although this was not found by [39]), might give indications for daytime dysfunction. 

In contrast, however, with previous research, most group differences were of a small size. The largest differences were observed in terms of the global score, sleep quality, and daytime dysfunction, especially between the LFN1 group and the CG. In this regard, it appears that it is not the LFN groups reporting little sleeping problems, but that the CG reports higher sleep difficulties than anticipated. The questionnaire developers provide scores from a sample of ‘good sleepers’, and of three clinical ‘bad sleeper’ groups ([68], see their Table 3) including individuals with depression, with disorders of initiating and maintaining sleep, and with disorders of excessive somnolence. On the one hand, the scores of our LFN participants partly resemble those of the ‘bad sleeper’ groups on all components except for daytime dysfunction. This would suggest that sleeping difficulties are elevated in the LFN groups. On the other hand, we observe that the CG in the present study scores twice as high on most variables and about ten times higher on ‘habitual sleep efficacy’ and ‘sleep medication’ compared to the scores of the ‘good sleeper’ group. Only the component of ‘sleep disturbances’ was comparable. Notably, the ‘good sleepers’ sample was not a community sample, and while this sample had a similar age distribution to our CG (24–83 years, average of 60 years), it consisted of 77% males. Thus, it seems possible that individuals with the demographic characteristics seen in our LFN groups and CG (higher age and a majority of higher educated females) might be prone to experiencing more sleeping problems regardless of LFN perceptions. Indeed, women and older individuals belong to the risk group for bad sleepers in The Netherlands; however, higher educated individuals do not belong to this risk group [76,77].

Finally, when considering the proportion of participants categorized as bad sleepers, the two LFN groups showed similarly high proportions (LFN1 = 75%, LFN2 = 79%) compared to a lower proportion in the CG (55%). This finding supports previous survey findings with similar, yet somewhat higher reported sleeping difficulty proportions of 54–77% [8], 82–89% [31], and 83% [6]. Notably, 90% of the overlapping sample in Erdélyi et al. [62] reported sleeping difficulties when asked on a multiple-choice question about LFN-related difficulties. 

In contrast to the proportion of bad sleepers in all of our groups, the proportion of individuals complaining about bad sleep in the Dutch general adult population is 24–29%, which is considerably lower [76]. Females and individuals between 50 and 64 years present with the highest proportion of bad sleep (i.e., 34%), but although many of our participants belong to this group, this proportion is still lower than the ones observed in our research. However, the current findings, especially those related to the LFN1 group, have to be treated carefully due to a large data loss of 23%. In conclusion, our results support the notions that sleeping difficulties are a highly relevant complaint in association with LFN perceptions, and that not all aspects of sleep are equally affected. Additionally, our findings suggest that the demographic characteristics of LFN complainants might be risk factors for sleeping difficulties. Further research into the interaction between sleeping risk factors and LFN perceptions on different aspects of sleeping difficulties is therefore needed.

#### 4.2.3. Fatigue

The two LFN groups reported similar degrees of fatigue symptoms and proportions of above average symptom reporting, which were considerably higher than for the CG. While previous research focusing on short-term fatigue in specific situations (such as work situations, during mental performance tasks, after sleep) gave first indications about the relevance of daily life fatigue related to LFN (e.g., [5,24,28,38,39,46]), our results suggest that long-term fatigue seems to be a relevant and common complaint in relation to everyday LFN perceptions. The proportions of above average symptom reporting observed in the two LFN groups (LFN1 = 57%, LFN2 = 58%) are in line with the significantly higher rated fatigue in individuals annoyed by LFN [13] and with the previously observed proportions of reported fatigue based on single-item questions (56% [6] and 59% [31]). Interestingly, 75% of the overlapping sample in Erdélyi et al. [62] reported fatigue when asked on a multiple-choice question about LFN-related difficulties. Notably, results derived from single-item questions and the current findings resulting from a structured questionnaire are not directly comparable. Overall, the current results suggest that more attention should be paid to long-term daily fatigue as well.

#### 4.2.4. Stress

Participants from both LFN groups reported more stress complaints and also presented with larger proportions of above average stress symptoms than the CG. This is in accordance with the notion that LFN is a background stressor and with previous research findings [5,48]. The proportion of individuals with above average stress complaints in our LFN1 group (62%) is in accordance with previous research, which found comparable proportions of participants feeling stressed in association with LFN (56% [32], 57% [6], and 31% high and 41% moderately stressed participants in [21]). 

However, more individuals in the LFN2 group presented with overall above average stress symptoms (78%) and reported considerably more daily stressors in their life compared to the LFN1 group. In contrast, the LFN1 group seems to experience stressors as worse and more intense than the LFN2 group. This discrepancy has not yet been reported in previous research. This might indicate that the type of daily stress difficulties could rely on factors on which our two LFN groups differ. Some examples of differing factors include the following: (1) their use of coping strategies, (2) their overall rated hindrance, or (3) their motivation to reach out for information and support as a result of our sampling method. 

Further, we found that the pattern of more reported stressors in the LFN2 group and the pattern of a higher reported stress intensity in the LFN1 group that was observed for all daily stress items, are also present for the dependent and independent items. The former refers to events that are dependent on the functioning of the participants, i.e., stressors being triggered by the person himself, such as “You didn’t keep a promise, or you didn’t stick to an agreement”. The latter refers to events that are not dependent on the functioning of the person, i.e., stressors triggered by forces outside the person, such as “You did not like certain developments or decisions in politics”. It is currently complicated to identify if complaints reported from LFN arise in terms of an external or internal stressor and if LFN complainants react differently to dependent or independent stressors, considering that in many cases the reported noise source cannot be measured or identified. The current data do not provide clear conclusions for this question; rather the data point towards the option that this might differ on an individual level. Rather than considering whether the stressor is internally or externally triggered to understand the perceived stress of LFN complainants, it could be more important to consider a related concept, the perceived control over a stressor. Lower levels of perceived control were suggested to be associated with higher stress levels after LFN exposure by Persson-Waye and colleagues [49]. Further, suggestions for the relevance of perceived control were also made in two treatment studies [21,32] and in conversations with affected individuals. However, perceived control still needs to be addressed in future scientific investigations. 

Overall, our results strengthen the evidence for subjective stress complaints related to LFN perceptions and suggest high levels of chronic stress. In contrast, the current body of research measuring situational and short-term stress with objective (e.g., cortisol level) measurements does not allow for clear conclusions yet. Since stress encompasses a wide range of definitions, types of measures, and a variety of possible symptoms, future research combining validated subjective stress measures with ecologically valid objective stress measurements for daily chronic stress would be valuable. A promising measure could be heart rate variability [78]. Further, the role of influencing factors, such as personality or coping strategies on stressors would be necessary. 

### 4.3. Summary LFN-Related Complaints

To summarize, our results indicate that individuals with reported LFN perceptions in their daily life report more complaints in all of the measured domains compared to individuals with no LFN perceptions. The most severe complaints were found to be cognitive difficulties and stress in terms of the largest observed group differences and the largest proportions of individuals with above average symptom reporting. Further, most individuals in the LFN groups showed above average complaints regarding sleeping difficulties. However, in our research a considerable proportion of the individuals with no LFN perceptions also showed above average sleeping difficulties. Moreover, we found that the LFN2 group, which was derived from a community sample and was therefore not specifically sampled for their LFN complaints, also reported considerable complaints on all measured domains. Considering that the LFN2 group reported LFN complaints to occur significantly less often (on average ‘sometimes’) and rated the hindrance experienced from LFN as significantly lower than the LFN1 group, our findings suggests that even individuals who report that they are less affected by LFN in their daily life seem to show daily life complaints. This lower complaint frequency and lower hindrance in the LFN2 group is in agreement with the fewer reported complaints in the domains of cognition and depression compared to the LFN1 group. However, both LFN groups showed similarly high symptom severity in terms of sleep and fatigue; and the LFN2 group reported even more daily life stressors. Therefore, it could be possible that sleeping difficulties, fatigue, and the frequency of stress are primary complaints occurring at low levels or early stages of LFN annoyance, while other complaints manifest later or as secondary complaints. Finally, it has to be considered that the complaint domains measured in this study can influence each other multidirectionally and are intertwined (e.g., sleep deprivation can lead to cognitive difficulties, stress can contribute to the development of mental illnesses). Thus, future research investigating the relationships between these complaints, but also the relation of complaints with the intensity and duration of LFN perceptions, or the role of influencing factors, such as annoyance, would be needed. 

### 4.4. Coping

Overall, the LFN1 group showed a tendency of scoring the highest on all coping strategies. This was followed by the LFN2 group and the CG, although these two latter groups showed similar scores on most strategies. This similarity between the LFN2 group and the CG might relate to the lower general hindrance from LFN reported by the LFN2 group compared to the LFN1 group. Since coping mechanisms represent a reaction to demanding or stressful situations, our findings could therefore indicate that individuals with lower general hindrances from LFN also have a lower need to use coping mechanisms. Future research would need to clarify which coping strategies are used specifically for LFN-related hindrances and investigate the relationship between the use of strategies and their success.

A striking finding of this study was the considerably higher use of support seeking strategies by the LFN1 group. One possible explanation could lie in our sampling method, since the LFN1 group was specifically gathered from a group of complainants that has previously reached out to a LFN foundation regarding their LFN experiences. Thus, the LFN1 group might have consisted of more LFN complainants with already higher support seeking tendencies. In comparison, the LFN2 group was derived from a community sample that did not specifically sign up for the research in order to report on LFN-related complaints. Another possible explanation could lie in the higher general hindrance observed in the LFN1 group compared to the LFN2 group. Accordingly, individuals with higher hindrances from LFN might also use more support seeking coping mechanisms. However, whether support seeking strategies are successful in dealing with LFN-related complaints has not been investigated. A third explanation could lie in the fact that LFN is often not perceived by others, which can make it especially difficult for affected individuals to receive understanding and support from others. Thus, support seeking behaviors might have developed as a reaction to perceived rejection. This might then also be in line with the suggestion by Leventhall and colleagues [32] that personal support poses a relevant coping component. 

Besides the frequent use of support seeking strategies by the LFN1 group, only few notable differences on the use of coping strategies were found between the three groups. First, the LFN1 group used more ‘active coping’ and ‘planning’ strategies compared to the other two groups. This aligns with previous findings that LFN complainants seem eager to try various actions to reduce their nuisance [8,60]. Second, the LFN1 group scored higher on ‘self-distraction’ compared to the other two groups. Notably, the questionnaire developers described ‘self-distraction’ (together with the support seeking mechanism ‘focus on venting emotions’) as a less useful coping strategy [52]. It is currently unclear whether these strategies are also less useful in the context of LFN, since ‘distraction’ has been previously suggested to be amongst the successful strategies for LFN complainants [60] and the questionnaire developers themselves pointed out that the usefulness of coping strategies highly depends on the context and the extent of their use. Finally, we observed that both LFN groups used more substances compared to the CG. This might partly relate to the higher sleep medication intake observed in both LFN groups (based on the PSQI questionnaire). 

Apart from this, we observed that all groups use a combination of problem-focused and emotion-focused strategies and that the coping strategies that the CG identified with the most were partly among the strategies that the LFN groups identified with the most. This alignment could indicate that LFN complainants and non-LFN complainants partly use the same main coping strategies. This finding is interesting insofar as it has been previously described that individuals with LFN complaints seem to have difficulties with coping and worry about their ability to cope with the noise [32]. The question remains whether LFN-related hindrances might require different coping strategies compared to other stressors. Further, all three groups scored similarly in terms of ‘acceptance’, although acceptance of the noise or its source has been described as difficult in previous research [32,60]. However, participants in one of those studies [32] were individuals whose complaints could not be previously resolved by environmental and healthcare professionals. Thus, it could be that ‘acceptance’ is a differentiating factor for a subgroup of individuals with high and chronic LFN complaints. 

In conclusion, our data suggest that first, partly similar strategies are used by all groups, second, that coping strategies are used the most by the LFN1 group, and third, that the LFN1 group uses considerably more support seeking coping strategies. However, the questionnaire did not ask for the specific purpose of the coping strategies. Accordingly, future research urgently needs to differentiate whether strategies are used for LFN-related or non LFN-related stressors, and which strategies are useful for reducing LFN-related hindrances. Also, further research investigating the relationship between the extent of hindrances from LFN and the use of coping mechanisms would be interesting, as well as the influence of experienced rejection from others on using support seeking strategies.

### 4.5. Strengths and Limitations

The recruitment methodologies used in the present study pose strengths and also limitations. The LFN1 group was recruited via a volunteer LFN organization. Therefore, it consists of participants that participated specifically because of their LFN-related complaints and that presumably perceive a higher burden from LFN compared to our LFN2 group. Accordingly, this might limit the representativeness of a heterogenous sample of LFN complainants from the general population. In this regard, the post hoc emergence of a group with LFN-related complaints of similar demographic characteristics from the comparison sample poses a major strength for the rigor of our study. Thereby, we were able to analyze data from two naturalistic groups from different referral contexts and include individuals with complaints with presumably lower levels of burden. This helped to reduce the risk of overestimating symptoms and dysfunction. Simultaneously, data from online research panels can hold the risk for participants filling in answers with little effort and therefore producing data with partly non-valid answer patterns. For this reason, respondents with static answer patterns were excluded and an additional analysis excluding participants that scored above the cut-offs for invalid answer patterns based on the BRIEF-A validity scales was conducted. This analysis showed no or only very small changes that had no meaningful effect on the outcomes. Eventually, it has to be considered that individuals with any diagnosis of a psychiatric or neurological disorder were excluded in the comparison samples (LFN2 and CG), while individuals with a psychiatric or neurological disorder with assumed low confounding effects on the outcome variables were retained in the LFN1 group. This sampling allowed us to assess the proportion of possible comorbid conditions that could occur in individuals with reported LFN perceptions and keep the group as naturalistic as possible. Simultaneously, it also ensured a comparison group with as little confounding factors as possible. However, this discrepancy has to be kept in mind when interpreting the results of this research.

Further, the subjective nature of this research poses both strengths and limitations. On the one hand, subjective measurements ensure cost-effective and ecologically valid measurements of daily life experiences in a large group of individuals with subjective LFN perceptions even without a successful sound measurement. On the other hand, this does not allow for causal conclusions between LFN exposure and subjective complaints. Together with this limitation to draw causal conclusions, it has to be considered that complaints, especially complex complaints such as depressive symptoms, are not exclusively predictable by single factors, such as LFN exposure. Rather, they also depend on other, non-acoustic factors, and are likely to be intertwined with other complaints. Further experimental studies including structured subjective measures and objective sound measurements will be needed in the future to determine the direct and indirect effects of LFN on daily life complaints and determine causes of LFN perceptions in the absence of a measurable noise. 

The large battery of questionnaires administered also poses both strengths and limitations. On the one hand, it allows us to gather a substantial amount of data, to address more complex research questions, and to make multidimensional investigations. On the other hand, with more variables, also the risk of false positives (Type I error) is rising. Further, participants had to fill out an extensive and long questionnaire. Therefore, also the chances for a respondent bias rise, where more participants with high motivation, energy, or available time might be included. Participants were under no time constraint to complete the questionnaires at once and had the freedom to stop and resume at a later point in time in case they got tired, bored, or lost motivation. Further, to limit the effect of such factors such as tiredness, boredom, and motivation, the LFN2 group and CG completed only half of the questionnaires and participants were financially rewarded. Participants in the LFN1 group completed the questionnaires on paper and could therefore integrate it in their daily routines to their liking. However, we cannot rule out that some of those factors had an impact on the completion of the questionnaires. Specifically, different motivations between the groups taking part in this research and the difference in questionnaire administration between the groups (online and on paper) have to be taken into account. Notably, the exclusion of participants with possible noncredible reports as determined by the BRIEF-A validity scales showed no marked effect on the outcomes.

Finally, the results of this study have to be treated with care in terms of the norm groups and cut-off scores that were used. The demographic characteristics of our target groups deviated in many instances from the norm groups determined by the questionnaire developers. Specifically, the norm groups of the BRIEF-A, BDI-II, and APLN used considerably younger samples and a more even gender distribution compared to the groups in this study. The cut-off scores might therefore have been more rigorous than with a more similar norm group. Also, the cut-off definitions for above average symptom reporting have to be treated carefully, since they were based on different definitions. First, for three questionnaires, a predefined cut-off score based on the raw score was used (BDI-II, PSQI, and FSS). Second, two cut-off scores were based on the percentile scores of a norm group (BRIEF-A and APLN). For the BRIEF-A, a cut-off score of 84% was used. In contrast, the authors of the manual of the APLN determined a less strict cut-off score of 80%. Finally, an 84% percentile cut-off score was based on our CG due to the lack of a fitting comparison group for the FLei. Accordingly, direct comparisons cannot be made.

## 5. Conclusions

The results of the current research indicate that individuals reporting LFN perceptions also report various daily life complaints in the domains of cognition, depressive symptoms, sleep, fatigue, and stress. Furthermore, it seems that also previously less investigated complaints, such as long-term fatigue, or memory and executive functions, are commonly reported and have to be paid attention to. Another finding of this study is that there are differences in the extent and frequency of complaints between different subgroups of LFN complainants and investigations of primary and secondary complaints are necessary in future research. Specifically, a group of individuals with overall lower perceived hindrances from LFN still reported similarly high sleeping difficulties and fatigue compared to a group with overall higher rated hindrances from LFN. While this former group reported less complaints in terms of cognition, depressive symptoms, and the intensity of stress, it reported more daily stressors in their life compared to the group with overall higher rated hindrances from LFN. Future research investigating the role of perceived control over experienced stressors could be promising in understanding LFN-related stress. An unexpected finding was that partly high complaints, especially in terms of sleep, have been observed in the CG as well. This could suggest that the demographic group of older, highly educated individuals with a majority of females might already be of higher risk for complaints regardless of LFN perceptions. Finally, our results in terms of coping mechanisms suggest that participants across all groups use a partly similar combination of multiple coping mechanisms in their daily life. However, it seems that individuals with high overall reported hindrances from LFN have a tendency to use all coping strategies more frequently, especially support seeking strategies. But in order to derive clear conclusions, more research about which strategies are used specifically for LFN-related hindrances and the success of the coping strategies would be necessary. To summarize, this research is the first to provide a cross-comparison between multiple daily life domains in such a large scale in naturalistic groups with reported LFN perceptions and with a battery of psychometrically validated measures. However, we highly recommend future experimental and replication studies to further investigate the relationship between both LFN exposure and LFN perceptions, and related complaints. 

## Figures and Tables

**Table 1 jcm-13-00935-t001:** Demographic characteristics of the LFN1, LFN2, and CG groups.

	LFN1 (*n* = 181)	LFN2-SA (*n* = 131)	LFN2-SB (*n* = 108)	CG-SA (*n* = 229)	CG-SB (*n* = 239)
Sex					
Females (%)	124 (68.5)	93 (71.0)	71 (65.7)	144 (62.9)	158 (66.1)
Education (%)					
Low ^a^	14 (7.7)	1 (0.8)	0 (0)	3 (1.3)	2 (0.8)
Middle	61 (33.7)	56 (42.7)	47 (43.5)	88 (38.4)	80 (33.5)
High	106 (58.6)	74 (56.5)	61 (56.5)	138 (60.3)	157 (65.7)
Marital status (%)					
Married ^b^	87 (48.1)	80 (61.1)	48 (44.4)	124 (54.1)	142 (59.4)
Unmarried ^c^ No partner Partner, living together Partner, not living together	64 (35.4) 34 (18.8) 26 (14.4) 4 (2.2)	31 (23.7) 13 (9.9) 15 (11.5) 3 (2.3)	35 (32.4) 15 (13.9) 16 (14.8) 4 (3.7)	54 (23.6) 24 (10.5) 22 (9.6) 8 (3.5)	55 (23.0) 32 (13.4) 20 (8.4) 3 (1.3)
Divorced	22 (12.2)	16 (12.2)	13 (12.0)	28 (12.2)	24 (10.0)
Widowed ^d^	6 (3.3)	4 (3.1)	12 (11.1)	23 (10.0)	17 (7.1)
	Mean ± SD (Range)				
Age in years ^e^	57.4 ± 11.3 (25–87)	54.5 ± 13.5 (18–86)	56.6 ± 13.0 (24–90)	62.7 ± 11.8 (26–89)	62.0 ± 11.5 (25–85)
Frequency of LFN complaints	3.2 ± 0.8 (1–4)	1.6 ± 0.9 (0–4)	1.7 ± 0.9 (0–4)	0.5 ± 0.5 (0–1)	0.4 ± 0.5 (0–1)
Extent of LFN hindrance	7.3 ± 2.3 (1–10)	4.8 ± 1.8 (1–10)	4.8 ± 1.8 (1–10)	1.3 ± 0.5 (1–2)	1.3 ± 0.5 (1–2)

Note: LFN1 = LFN group recruited via LFN foundation, LFN2 = LFN group recruited via online panel, SA = subsample A filling out questionnaires regarding cognition and depressive symptoms, SB = subsample B filling out questionnaires regarding sleep, fatigue, stress, and coping, CG = Comparison group recruited via online panel, ^a^ significant group difference between the LFN1 and all other groups based on Chi-square tests. ^b^ Includes marriage and registered partnership. Significant group differences between LFN1 and LFN2-SA, LFN1 and CG-SB, LFN2-SA and LFN2-SB, and LFN2-SB and CG-SB based on Chi-square tests. ^c^ Significant group difference between LFN1 and LFN2-SA, LFN1 and CG-SA, and LFN1 and CG-SB based on Chi-square tests. ^d^ Significant group difference between LFN1 and LFN2-SB, LFN1 and CG-SA, LFN2-SA and LFN2-SB, and LFN2-SA and CG-SA based on Chi-square tests. ^e^ Significant difference between LFN1 and CG-SA, LFN1 and CG-SB, LFN2-SA and CG-SA, and LFN2-SB and CG-SB based on Mann–Whitney U tests.

## Data Availability

An anonymized data set used for this study can be accessed from the corresponding author upon reasonable request.

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
