# Peer review of "Subjective Complaints and Coping Strategies of Individuals with Reported Low-Frequency Noise Perceptions"

_jcm, 2024, doi:10.3390/jcm13040935_

Round 1

Reviewer 1 Report

Comments and Suggestions for Authors

The study investigates subjective complaints related to Low-Frequency Noise (LFN) and exploring coping mechanisms used in daily life. Overall, the content is clear and detailed, providing a comprehensive overview of the research.

 Here are a few suggestions to enhance clarity and readability.

Abstract

 Clearly state the key findings at the end of the paragraph. Summarize the main outcomes and their implications. This will provide readers with a quick understanding of the study's results

Introduction

Consider rephrasing the sentence about the Dutch Institute for Public Health's definition to improve readability.

Ensure consistency in terminology, particularly when referring to Low-Frequency Noise. For example, you mention "LFN perceptions," and later, you use "LFN exposure." Sticking to one term throughout will help maintain clarity.

Clarify that the prevalence estimates mentioned (2% to 34% with a pooled prevalence of 10.5%) are for the proportion of the population perceiving LFN, not necessarily being annoyed by it

Consider restructuring the last part of the paragraph for better flow. Organize information about the various complaints associated with LFN perceptions into a more cohesive structure, such as categorizing them under physical, psychological, cognitive, and social domains.

Consider using "LFN" consistently throughout the text instead of variations like "Low-Frequency Noise."

Consider breaking down long sentences into shorter ones for improved readability.

Ensure consistency in the use of tenses. For instance, you switch between present and past tense in some sections. Maintaining a consistent tense throughout the paragraph would enhance coherence.

Consider as a limitation that depressive symptomatology may not be solely predictable by LFN noise exposure and but may depend on other non-acoustic factors.

Reviewer 2 Report

Comments and Suggestions for Authors

Thank you very much for the opportunity to review the manuscript. The research is interesting and relates to an important issue.

1) The introduction is well-written and provides important information. However, there is room to shorten it and perhaps do more integration between the variables. The data is really interesting but because it is so long I feel it makes the text difficult to follow.

2) Research aims and questions are clear.

3) The Authors stated: "In order to control for alpha error growth in multiple testing, a strict significance level of p < .01 was applied and interpretations additionally based on effect sizes". 

I think that the Authors should explain why they applied a significance level of p < .01? It may be correct but they should explain why.

4) How did the subjects deal with such a large amount of items in the questionnaires? (fatigue?? boredom?? motivation??).

5) I think there is room for consistency in the sequence of presentation of things, the introduction started with sleep-fatigue-cognition-depression-stress and then the discussion started with cognitive functioning-depressive symptoms-sleep. 

6) I think that multiple tests and items may be both a strength and on the other hand limitation of this study. 

7) The results are interesting but it was very hard to follow all the results.

8)  The Authors should cite more new references.

In sum, I think it is a very important study and interesting topic, however, the Authors should improve the manuscript and revise it. 

Round 2

Reviewer 2 Report

Comments and Suggestions for Authors

Well done. I think the writers have addressed all my comments and corrected accordingly.